# Negotiating Awareness: Dutch Midwives’ Experiences of Noninvasive Prenatal Screening Counseling

**DOI:** 10.3390/ijerph192215283

**Published:** 2022-11-18

**Authors:** Marieke de Vries, Danique Oostdijk, Kim G. T. Janssen, Raymond de Vries, José Sanders

**Affiliations:** 1Institute for Computing and Information Sciences, Radboud University Nijmegen, 6525 EC Nijmegen, The Netherlands; 2Communication and Information Sciences, Radboud University Nijmegen, 6525 XZ Nijmegen, The Netherlands; 3Medical Biology, Radboud University Nijmegen, 6525 XZ Nijmegen, The Netherlands; 4Center of Bioethics and Social Sciences in Medicine, University of Michigan, Ann Arbor, MI 48109, USA; 5Centre for Language Studies, Radboud University Nijmegen, 6500 HD Nijmegen, The Netherlands

**Keywords:** noninvasive prenatal screening (NIPS), midwifery communication, decision making, counseling, informed consent

## Abstract

Background: Discussion of the topic of noninvasive prenatal screening (NIPS) has become a standard part of Dutch maternity care practice. This means that pregnant women who are contemplating NIPS can receive counseling from their midwife or obstetrician. The aim of this study is to understand the communicative practices and decision-making principles regarding first-tier use of NIPS, as experienced by Dutch midwives. Methods: Qualitative analysis of in-depth interviews with Dutch midwives (n = 10) exploring their conversations about NIPS counseling and decision making. Results: Midwives value the autonomy of women in decisions on NIPS. They consider it a midwifery task to assess women’s awareness of the risks and implications of using or not using this mode of screening. The optimal level of awareness may differ between women and midwives, creating novel challenges for informed decision making in midwifery communication. Key conclusions and implications for practice: Negotiating awareness about NIPS in individual women is a relatively new and complex midwifery task in need of counseling time and skill. NIPS practices call for a reflection on midwifery values in the context of integrated maternity care.

## 1. Introduction

In the Netherlands, integrated maternity care is provided primarily by midwives and consists of prenatal, intrapartum, and postnatal care [1]. During the first trimester of a woman’s pregnancy, intake and counseling about noninvasive prenatal screening (NIPS) take place [2]. NIPS tests are used to assess whether the unborn child may have Down, Edwards, or Patau syndromes (also known as trisomy 21, 18, and 13, respectively). As the name implies, NIPS tests are noninvasive, and as of 2017 they are offered to all pregnant women in the Netherlands [2,3,4,5]. Because of the preference-sensitive nature of prenatal screening, it is important that NIPS does not become routine. The decision to use NIPS should be based on a well-informed and value-concordant decision by pregnant women [2,3,6,7,8,9,10]. Therefore, information about NIPS is included in the protocol and scheduled as a conversation topic during consultation to ensure that women can make a well-informed decision [6,11].

Most Dutch women who want to be informed about prenatal screening are counseled by midwives, whose task it is to safeguard an informed and deliberate decision-making process for pregnant women [3]. Considering first-tier use of NIPS, midwives guide women to five decision points: whether they want to know about, and actively process information on, NIPS (the right not to know); whether they want counseling on NIPS; whether they want to participate in NIPS; if so, which particular test they opt for; and whether they opt for being informed about potential additional findings (i.e., all chromosome abnormalities other than trisomy 21, 18, and 13; [12]). Ensuring informed and deliberate decision making in these sequential decision tasks in a balanced way is a substantial task. When done well, it results in positive healthcare communication, healthy relationships between care providers and receivers, and good outcomes for pregnant women [13]. In a survey among Dutch midwives about their role as counselors, they were found to give information about NIPS and to be confident in their abilities to counsel about NIPS [14]. Since 2017, NIPS became generally accessible as part of the implementation of the TRIDENT-2 studies, implying that Dutch midwives are trained to counsel pregnant women about first-tier use of this type of test [15]. Anticipating the nationwide implementation of NIPS, Dutch midwives expected that counseling for NIPS as a first screening test would become more important and also that it would be relatively easy [13]. An important question is, therefore, what kinds of experiences and reflections Dutch midwives in fact have regarding counseling about NIPS. In this study, we examined how midwives talk about the ways they safeguard informed and deliberate decision making in first-tier use of NIPS consultations and on the outcomes of NIPS decision making for themselves and for the pregnant women they provide care to.

## 2. Materials and Methods

### 2.1. Participants

Potential participants were informed in advance that the goal of the study was to gain insight into the experiences of midwives with communication about NIPS in midwifery practice. They were informed in advance and again on the day of the interview (before and after the session) about study procedures, audiotaping, anonymity, and the possibility to withdraw or correct data. Written informed consent was obtained prior to study inclusion. Participants were recruited via the researchers’ personal network, by contacting midwifery practices, and via referrals from other participants. Participants were not paid for participation. About half of the midwifes who were contacted agreed to participate; the other half refused to participate because of a lack of time. Enrolment continued until no new issues arose anymore, which was after eight interviews were held. After preliminary analysis of the data generated by these interviews, we held interviews with two additional respondents as a saturation check [16]. All eight participants in the primary data collection round were certified Dutch female midwives working in group practices in the provinces of Gelderland or Utrecht in the Netherlands. We chose to recruit throughout the larger areas of Gelderland and Utrecht, allowing us to cover both rural and urban contexts, traditionally catholic and traditionally protestant areas, and to include midwives from larger and smaller practices in various SES contexts. In addition, as part of our purposive sampling, we made sure that all midwives who participated had received training in counseling for first-tier use of NIPS. The midwives had an average age of 37.4 years, with the youngest midwife being 27 years old and the oldest participant being 63 years old. The two saturation check interviews were conducted with two certified Dutch midwives, highly experienced in midwifery, including NIPS counseling, and now working as coaches and trainers for other midwives and midwifery students. These respondents, aged between 50 and 55, had a broad contemplative overview of Dutch midwifery practices; one was stationed in the center of the Netherlands and one in the rural north of the Netherlands.

### 2.2. Procedure

The Ethics Assessment Committee for the Humanities of Radboud University approved this research project (ETC-GW number 2019–7072). Interviews were held while midwives were in their practice or home. The saturation check interviews were conducted via videoconferencing due to COVID-19 restrictions. Interviews were conducted by a female assistant (master’s student; Danique Oostdijk, first eight interviews) or a senior female researcher (full professor, with ample experience in conducting interviews) and a female assistant (José Sanders, full professor, PhD and Kim Janssen, master’s student: two saturation check interviews). The highly experienced full professor supervised Danique Oostdijk and Kim Janssen in conducting interviews. None of the interviewers was personally acquainted with the participants. No other persons were present during the interviews. The interviewees were informed about the authors’ reasons for conducting research. The interviewers explained that they were interested in communication in healthcare and, more specifically, in communication in maternity care. José Sanders explained that she had conducted research on this topic before. Interviews were audiotaped and interviewers also made some field notes during the interviews. Applying a semi-structured method, the interviewer posed neutral questions using neutral language. This entails that she avoided statements that contained any type of judgement and instead used only content-neutral speech to elicit, clarify, expand, or contrast participant responses [17]. More specifically, the interview method used avoided evaluative content expressions (such as ‘problem’ or ‘bad’; ‘normal’ or ‘strange’) and instead invited clarifications without offering interpretations (i.e., only posing questions such as ‘what happened next?’; ‘if that happens, what do you do?’; ‘upon hearing this, how was that for you?’). This so-called clean language procedure allows participants to choose their own words and freely react to others’ words, which ensured that resulting conceptualizations and evaluations were generated by participants themselves rather than partly elicited by the moderator [18,19]. The following topics were introduced and discussed: midwifery procedures for informing and counseling about NIPS; needs of pregnant women (and their partners) before and during counseling about NIPS; processes of decision making about NIPS for women; and experiences of and challenges for midwives during informing, counseling, and decision making about NIPS. Interviews lasted 60–90 min.

### 2.3. Analyses

Analyses of the transcripts of the first eight interviews were performed using a bottom-up approach based on grounded theory for identifying communicative themes [20]. The analysts collaborated in analyzing the different topics addressed by the participants. Transcripts were read repeatedly to achieve data familiarity. The transcripts were coded in the program ATLAS.ti using the matrix of themes found in the interview data analysis. Interpretive codes were created for categories in which participants discussed experiences of NIPS communication during midwifery consultations. Relevant parts were also labeled and grouped into categories by axial coding, focusing on specific types of verbalizations [21]. After identifying a core category that indicated a distinctive theme, the transcripts were re-examined by a second coder to validate the theme’s salience by searching for additional indications and possible counterindications. This process was repeated several times and discussed between analysts in order to arrive at an exhaustive set of relevant themes that covered the range of participants’ significant experiences in their communication with women about NIPS. In the saturation check interviews, we discussed results and no new themes emerged, supporting the validity and completeness of our findings. We used the consolidated criteria for reporting qualitative research (COREQ) [22] to check the reporting of our study according to the COREQ criteria; see Appendix A for the completed checklist. Respondents were given the opportunity to add to or withdraw anything stated in the interview, but none made use of the opportunity. To this end, they received the contact details of the interviewers. None of the interviewees contacted the researchers after the interviews. No repeat interviews were conducted. 

## 3. Results

Based on our analysis, we first describe what midwives reported regarding the counseling procedures. Subsequently, we identify three major themes they talked about: awareness, autonomy, and closeness. In the translated excerpts below, we offer exemplary cases of each theme.

### 3.1. Counseling Procedures

Midwives reported that most practices provide two consultations to counsel women about NIPS. The first typically takes place in weeks 7–8 and often is combined with the viability ultrasound. The second takes place in weeks 9–12 and is typically combined with the dating scan. NIPS tests can be requested from week 11. In case a practice starts with an intake, the first appointment includes bringing up the topic of NIPS and giving the women more information on paper in order to prepare for the counseling session. Alternatively, women may be asked to prepare for NIPS counseling in the welcoming email or letter. Typically, a reference to an online information source (for instance, www.pns.nl) is included. In such cases, the first appointment consists of (part of the) counseling, and the second consultation is used to answer any final questions women may have. Prior to providing additional information during consultation, midwives are obliged to ask if women would like to receive more information about NIPS, because women have the right not to know [23].

*I then ask them whether they’ve had a look at the website, whether they’ve already looked up more information on it themselves and whether they want any clarification, because they obviously have the right not-to-know, so that’s what I start with.* Respondent 4

(Dutch: Dan vraag ik aan ze of ze naar die website hebben gekeken en of ze daar al wat meer informatie zelf over hadden opgezocht. Of ze verduidelijking willen, want ze hebben uiteraard het recht op niet weten, dus daar begin ik mee.)

Part of the information about NIPS has sometimes already been provided by an assistant via the telephone when the intake is scheduled; some midwives feel that the right not to know is then compromised. In addition, midwives may avoid mentioning NIPS in the first consultation because they feel that discussing potential abnormalities does not seem appropriate in early pregnancy, as the experience of being pregnant should be considered first, and also because the risk of a miscarriage is still present. 

*Back in the day, we used to do the counseling during the intake interview, but [...] [that] feels very ambivalent, because people come in like hip hip hooray, I’m pregnant and then you have to tell them now let’s talk about possible abnormalities as well.* Respondent 3

(Dutch: Vroeger deden we dat eigenlijk bij de intake [...] [maar dat] was heel erg dubbel, want dan komen mensen met hieperdepiep ik ben zwanger, en om dan te zeggen laten we het nou dan ook even over de afwijkingen hebben.)

The time for counseling NIPS is generally limited to 30 min; in some cases, this is insufficient. If this is the case, a midwife typically tries to give the essential information about the tests so that she can skip part of her explanation and women are still able to decide. Alternatively, when women are not prepared, a midwife may decide to postpone the counseling session altogether. 

*Sometimes, I get people who haven’t read the brochures at all. I can then make a different decision as to what to do. […] Sometimes, I say if you haven’t discussed this with each other or read anything about it, I don’t have enough time to give sufficient explanation about all three tests in half an hour, in addition to discussing both the technical and ethical aspects to it. So then I occasionally ask them to come back another time and do the preparations first.* Respondent 7

(Dutch: Soms ook wel mensen die echt helemaal niet de boekjes gelezen hebben, daar maak ik soms een verschillende keuze in wat ik kan doen. [...] Ja soms zeg ik dan als jullie het er niet samen over gehad hebben en ook niet gelezen, dan heb ik eigenlijk ook niet genoeg tijd om in een half uur van al die drie onderzoeken voldoende uit te leggen, zowel als het ethische aspect met hun te bespreken, als de technische kant van het verhaal. Dus dan vraag ik soms toch nog om opnieuw terug te komen en toch nog wel even die voorbereiding te doen.)

### 3.2. Theme 1: Awareness

Midwives noted that it is important that women are aware of what NIPS is and what it might entail. They think it is essential that women (and their partners) prepare themselves, as it is not an easy subject and there is a lot of information available to parents. Specifically, women should be aware that participating in NIPS is not something they should agree to as if it is part of standard care; rather, that they should really think their choices through, because participating in NIPS can bring tension and uncertainty. Midwives said it is essential for women to think beforehand about what they would decide in each of the possible test outcomes.

*I think that they shouldn’t choose to take the test first and then later decide what to do with the results. I think that, if you want to get tested in order to be prepared for [a child with certain issues], that you may decide differently, from when you get tested because you have decided that you will terminate your pregnancy [in case of a bad test result].* Respondent 5

(Dutch: Ik vind dat ze niet eerst voor onderzoek moeten gaan en dan moeten bedenken wat ze ermee willen. Ik vind dat als jij onderzoeken wil laten doen ter voorbereiding op [een kind met bepaalde beperkingen], dat je misschien andere keuzes maakt, dan dat je doet op het moment dat je ervoor kiest om de zwangerschap af te laten breken [bij een slecht testresultaat].)

Our interviewees reported that sometimes women have a shallow awareness regarding NIPS and do not have a clear point of reference from which to make decisions. 

*People who don’t have anything to hold onto, how to say, they are stuck in a superficiality of some sort and are like, my sister took it, so I’ll take it too, or: no, we don’t do that. They bury their head in the sand and are like we’ll see. People take the test with the idea that it’s just a box to be ticked and, if they’re forced [to think about it later] they’re like we’ll see*. Respondent 9

(Dutch: Want mensen die helemaal geen houvast hebben of ten minste, ja hoe moet ik dat zeggen, die blijven in een bepaalde oppervlakkigheid op een bepaalde manier zitten en oh m’n zus heeft het gedaan dus ik doe het ook, of nee, dat wordt bij ons niet gedaan. Kop in het zand, ja dat zie ik dan wel. Mensen doen een test met het idee van je kunt het afvinken. En als ze dan gedwongen worden: “ja dat zie ik dan wel”.)

Midwives mentioned that during counseling they provide and check information. They assess if women are aware of the risks, as some women, in their experience, simply assume they will receive good news regarding the test result and say that they will worry only if this is not the case. The respondents think this may be a protection mechanism: a feeling during early pregnancy that prevents them from talking about their pregnancy, or questioning it, let alone thinking about terminating it.

*People also very often say something like I’m still young and it doesn’t run in the family. And every time I will say: I’m going to play devil’s advocate, but it’s only hereditary for a small percentage and, for the rest, it’s just coincidence of nature. [They will say:] Ooh, really. You can feel that sometimes it’s just a protection mechanism, like, it won’t happen to me*. Respondent 6

(Dutch: Andere mensen geven ook heel vaak een antwoord van: ja ik ben nog jong en het komt niet in de familie voor. Ja, en iedere keer dan zeg ik weer opnieuw: even advocaat van de duivel, maar slechts een heel klein percentage is erfelijk bepaald en de rest is ook gewoon toeval van de natuur. [Dan zeggen ze:] “Ooh ja”. Je merkt dat het soms een beschermingsmechanisme is van: het overkomt mij niet.)

Midwives said that during counseling it happens that women decline information about NIPS initially and seem to try to avoid the information. In such cases, after the midwife has offered some information, some women in fact do want to learn more. Possibly, these women do not want to reflect on their options regarding NIPS, since there is a very good chance that they will receive a test result that does not confront them with the matter of terminating their pregnancy anyway. Such resistance against knowing more about their options is, in the experience of midwives, a way for pregnant women to protect their child. 

*There are quite a few women who, at an early stage of pregnancy, suddenly feel what a strong maternal instinct they have developed to protect their child*. Respondent 10

(Dutch: Er zijn best veel vrouwen bij wie al in een vroeg stadium in de zwangerschap ze ineens voelen wat een sterk ontwikkeld moederinstinct ze hebben om dat kind te beschermen.)

In addition, midwives noticed that with some women it is difficult to raise their awareness about certain aspects of NIPS: test results can be difficult for the women to interpret and can cause anxiety.

*One person says, I just want to know what there is to know and then I’ll see, and another says, I’m already an anxious and insecure person so I think that [testing] will only make me more anxious. So it differs a lot per person.* Respondent 5

(Dutch: De ene zegt, ik wil gewoon weten wat ik kan weten en dan zie ik het daarna wel en de ander zegt, ik ben altijd al meer onrustig en onzeker van mezelf dus ik denk dat het mij alleen maar onrust gaat geven. Dus dat is heel verschillend per persoon.)

The need for being unconcerned can also surface in another way: some women experience an inability to choose. According to respondents, some women do not like the fact that they have to make decisions regarding prenatal screening so early in their pregnancy and would prefer not to be aware of their options. 

*It varies. There are people who are prepared for the fact that this is coming at them and for these questions and the opportunity to make a choice in this matter [...] However, there are also people, especially women, who find it very difficult to be given this choice. What also counts is the fact that this is brought up very early in the process. Some people, after having tried for very a long time to get pregnant…don’t want to think about these things, either not at all or not yet*. Respondent 7

(Dutch: Dat is wel wisselend. Er zijn namelijk mensen die dus voorbereid zijn op het feit dat dit eraan komt en de vraagstukken en dat ze daar een keuze in mogen maken [...]. Maar er zijn ook wel, met name vrouwen, die het heel erg lastig vinden dat ze een keuze krijgen. En dan speelt ook nog wel een rol dat je dat heel erg in het begin aankaart, dat mensen [...] na heel lang proberen zwanger te worden eigenlijk over deze dingen helemaal niet…überhaupt niet of nog niet willen nadenken.)

### 3.3. Theme 2: Autonomous Choice

Midwives emphasized the need to respect women’s autonomy in choices regarding NIPS. Midwives therefore make an effort to offer counseling that is value-free, and that is something to be learned by experience. 

*Because it’s their choice. I’m a bit fussy about that. [...] I find that, when I hear interns practice counseling, I am like, don’t do it, talk about the person herself*. Respondent 2

(Dutch: Want het is hun keuze. Daar ben ik wel een beetje pietluttig in. [...] Dat heb ik ook als ik stagiaires hoor counselen dat ik denk, niet doen, heb het over de persoon zelf.)

Midwives believe it is essential to find out if the choice a woman makes is really what she wants and is concordant with her values. To this end, they take extra time to find out what the values, preferences, and considerations of women are. Midwives reported that women are also influenced by developments in healthcare and view testing as an unalloyed good. Moreover, they respect women’s individuality and try to adapt to their needs and preferences by offering a postponement to think about options. 

*Sometimes, I mostly give them time. I indicate that they don’t have to make a decision right now*. Respondent 7

(Dutch: Soms geef ik ze vooral tijd. Ik benoem dat ze niet nu al een keuze hoeven te maken.)

When women still have questions before they can choose, midwives tell them to call, or they schedule an extra consultation. When women come in well prepared, midwives have more opportunity to explore the values and considerations of the women and their partners. Due to their preparation—by reading information or by talking to people in their personal environment—many women understand that they will need to make choices regarding NIPS. If needed, midwives help the women they counsel with their decision, for example, by referring to an online decision aid. Furthermore, midwives help by exploring differences between options and advising women to talk with people in their personal environment. 

*Then we try to give them some more tools for making their decision, so we have them fill out the decision aid […] again and we make a list of pros and cons together with them*. Respondent 5

(Dutch: Dan proberen we ze wat meer handvatten te geven, dus nog een keer die keuzehulp […] te laten invullen. En ook gewoon door samen dingen op een rijtje te zetten met de voor- en nadelen.)

#### Subtheme ad 2: Barriers to Women’s Autonomous Choice

Midwives see that, while the majority of women come prepared to the counseling session by reading the website or other information sources, there is a group of people that choose not to read the information because they do not want to, due to religious reasons, for example. In addition, some women have not read the information due to a language barrier or lower education levels. When women with a language barrier bring along someone to translate for them, midwives do not always know if the translation is correct and complete. Midwives reported that they find this difficult because they cannot check the exact message the women receive and whether it is communicated value-free. In such a situation, it also is difficult to know if the ultimate decision really is the woman’s autonomous choice, matching her own views and values. 

*I don’t know how the message is received by them, whether it’s really their own choice and whether we’re really trying to figure out what’s best for her or whether there might be a hitch somewhere after all*. Respondent 1

(Dutch: Omdat ik niet weet hoe die boodschap dan aankomt, en of het wel echt een eigen keuze is. En of we echt kijken wat er voor haar past, of dat er misschien toch ergens een kink in de kabel zit.)

When encountering women with choosing barriers, midwives go to great lengths to prepare women for choosing.

*People with a lower level of education for instance…they don’t read everything right away and, with immigrants, where there’s a language barrier, you’ll see that you discuss these things more elaborately during the first consultation. I also always try to provide them with a link to information in the language that they speak, but you’ll often see that it isn’t happening. And then we’ll even print the information so people can take it home*. Respondent 8

(Dutch: Wat lager opgeleide mensen bijvoorbeeld, die… nou vind ik niet altijd dat die het helemaal gelijk gaan doorlezen. En buitenlandse mensen. Daar zie je dan van met de taalbarrière dat je dat meer bespreekt op de eerste controle en ik probeer dan ook altijd al de link mee te geven van de taal die ze spreken, maar dat zie je toch dat dat minder gebeurt. En dan printen we het vaak hier voor de mensen uit nog, zodat ze het mee naar huis nemen.)

### 3.4. Theme 3: Closeness

Midwives report that they try to imagine what testing in early pregnancy means for women, and they report how they talk with women about this during counseling. The next excerpt illustrates such an attempt to reach empathetic nearness.

*You could have a very pleasant conversation with somebody at twelve weeks of pregnancy about everything that’s coming at them, because it’s quite something, and now you have to start the whole trajectory with a very impersonal approach regarding a possible termination of that pregnancy. […] So I try to sugarcoat it for them by saying you know, this sucks. I just say it, this sucks, and I explicitly mention that, when you’re having a planned pregnancy and you want to have a baby very much—now we are going think about what if there’s something wrong with that baby*. Respondent 10

(Dutch: Daar waar je met 12 weken iemand die in blijde verwachting is een fijn gesprek hebt over wat komt er zoal op je af, want het is nogal wat er op je af komt, en dat je nu eigenlijk moet beginnen het hele traject met een vrij zakelijk iets wat gaat over afbreken van die zwangerschap. [...] Ik probeer die pil dan dus ook wel te vergulden voor mensen, dat ik dus inderdaad zeg van weet je dit is waardeloos. Ik zeg het ook gewoon, dit is waardeloos en ik benoem dit ook gewoon precies zo van dit is een, als het gaat er om een gewenste zwangerschap, je wil heel graag een baby en dan gaan we dus nadenken over wat nou als er met die baby wat loos is.)

This illustrates how midwives try to verbalize how women may feel about testing, and how they empathize with this. Generally, midwives value being close to women: their aim as a midwife is to reach optimal socio-psychological proximity to the women they counsel. Given this desire, midwives think it is important to level with the women when offering counseling about NIPS. One respondent expressed that she values the aspect of intimacy in her job. She went on to describe counseling as something intimate, concluding that counseling is really part of, and fits well within, her profession. 

*You’re really making important life choices. You’re talking about something very intimate. Of course, that goes for my whole profession*. Respondent 2

(Dutch: Dan ga je echt over levenskeuzes. Ja, dat is eigenlijk iets heel intiems waar je het over hebt hè. Wat natuurlijk mijn hele vak wel is.)

Yet, respecting the choices of women should not be compromised. Women may make decisions that are not based on midwifery values. One of the respondents indicated that this is something she experiences as difficult. 

*So if you don’t relate to someone’s point of reference at the moment, you won’t reach them. So you can only start counselling if somebody is receptive to that at the moment. […] You feel as if you’re going too far. You can’t do that and, if you do, people won’t want to see you anymore, because you’re too intrusive. You don’t want that because you want to be as close as possible to them. So I’ve really learned not to do that.* Respondent 9

(Dutch: Dus als je niet aansluit bij waar iemand op zo’n moment is, dan bereik je iemand ook niet. Dus je kunt pas gaan counselen ook als iemand daar op dat moment open voor staat. […] Ja je voelt dat je te ver gaat zeg maar. Dat mag niet en als je dat dan doet, dan word je…Ik zou bijna zeggen dan willen mensen je niet meer zien, want dan word je te indringend. En dat vinden mensen eng en gevaarlijk en dat komt te dichtbij. En dat wil je niet, want je wil juist die maximale nabijheid. Dus ik heb echt wel geleerd om dat gewoon niet te doen.)

This excerpt illustrates how midwives balance proximity with women, depending on women’s preferences and their need for awareness. This may challenge their own values of professional information giving. For instance, midwives noticed that people’s perspectives on Down syndrome are frequently influenced by contextual factors, such as media information, religious beliefs, or the experiences of friends or family members. These factors can lead to incorrect assumptions on the part of women that are not easy to correct. 

*If people don’t understand it and have prejudices or assumptions beforehand that are incorrect and hard to correct, I sometimes find it difficult*. Respondent 1

(Dutch: Als mensen het niet goed begrijpen en van tevoren vooroordelen of aannames hebben die echt niet kloppen, die ik ook moeilijk recht krijg, dan vind ik het soms wel lastig.)

Midwives also reported that, if needed, they redirect the decision to women by pulling back; they make sure it is the women who decide by actively placing the decision before the women, away from themselves, and also away from the women’s personal environment. 

*I’ll call them back, like “nobody can do this in your place, because it’s something that you have to do yourselves. It is your life, you’ll have to go and do it”.* Respondent 6

(Dutch: Dan haal ik ze wel even terug van: “Niemand kan die in jullie plek, want het is iets wat jullie zelf moeten doen. Het is jullie leven, jullie moeten het gaan doen”.)

If necessary, midwives also redirect by pushing back; according to the respondents it frequently occurs that women ask midwives what they would do or have done themselves regarding NIPS. Most midwives experience this as a difficult question. They respond by saying that it does not matter what they would do or have done themselves and that it is the women’s choice. They also respond by asking what the women’s view is about NIPS. 

*So then I tell women yes, other people also ask that question, but I can’t make that decision for you, because I can’t raise your children either, so it’s a decision that must suit you*. Respondent 7

(Dutch: Dan benoem ik dat ook aan mensen: “dat wordt mij vaker gevraagd, maar ik kan die keuze voor jou niet maken, want ik kan ook jullie kinderen niet groot brengen. Dus is een keuze die bij jullie moet passen.)

## 4. Discussion

Midwives value women’s autonomy in their decision making and also in their preferred level of awareness about the consequences and risks regarding NIPS. Finding the optimal level of awareness in each individual woman is a complex task that requires dialogue and reflection. 

### 4.1. Counseling to Raise Awareness

Securing consultation time for dialogue and reflection is challenging when women have many questions, when it is their first pregnancy, when women find it difficult to decide whether or not to test, or when a pregnant woman and her partner disagree about testing. Midwives feel that it may be difficult to provide a sufficient explanation when women are not prepared, unaware, or unwilling to decide. This is significant, since it is important to midwives that women are aware (or think) [24] and consider their choice options carefully, not assuming the test is just a standard thing to do. While women may consciously or unconsciously choose to be unconcerned, midwives see their role as guiding women into reflection about what decisions best suit their life and values, while simultaneously avoiding making the NIPS decision for them. Women may actively avoid knowledge about NIPS and the potential psychological impact of invasive diagnostic tests. Some midwives gently confront a woman when they feel she cannot, or prefers not to, face the consequences of a test [25]. For these midwives, NIPS counseling is a strenuous task. It is understandable that midwives may, for principled or pragmatic reasons, develop a routine of limiting themselves to providing information and making sure a decision is made. In such cases, their counseling may not quite attain the level of informed consent [26]. 

### 4.2. “Being with the Woman” While Acting in Accordance with the Protocol

The midwifery profession is challenged to move from “being with the woman”—following and guiding the physiological process of pregnancy and birth—to a medical, “evidence-based”, more distanced role of advising and intervening as required by a protocol. Guidelines and protocols are designed to ensure that pregnant women and their unborn babies are offered maternity care in line with the best available medical evidence, primarily anchored in a risk-based approach promoting safety and risk management [27]. Healthcare providers increasingly encounter women in their care practices who request less, or different, care and interventions than recommended by their healthcare providers [28]. Often, these women are making choices that reflect their personal values, circumstances, and needs. These preferences may not contradict medical guidelines, but they are expressions of aspects of care not typically captured in medical trials, such as the continuity of the care provider, autonomy, and respectful maternity care [29]. These values have been shown to be of crucial importance to having a positive experience with maternity care [28]. In the context of NIPS counseling, these findings underline the importance of the traditional role of midwives: “being with the woman”, building a trusting relationship that considers the personal needs, experiences, and preferences of women. As stressed by the World Health Organisation, a positive experience with maternity care has important implications for the short- and long-term well-being of the woman, her baby, and her whole family [30].

### 4.3. Value Stretching and Reflection

The current transition to an integrated maternity care system in the Netherlands includes asking midwives to counsel women about their choices for NIPS. Our study illustrates that for midwives, integration of this counseling in their routine is time-consuming and “value stretching”, especially if it is to be performed in a tailored-to-women way. As our study illustrates, midwives observe that some women tend to approach NIPS as a routine procedure that does not involve making a deliberate decision. This is in line with Cernat et al.’s review [31] study findings of women often describing NIPS tests as easy or just another blood test, which highlights threats to informed decision making such as routinization or a pressure to test. Raising these women’s awareness to achieve informed consent may be considered as a minimal strategy. It depends on midwives’ counseling procedures, such as the time available for counseling, but also on midwives’ values, such as being empathetic with women and respecting their autonomous choices, whether deliberate decision making is strived for. When women consciously or unconsciously wish to stay unaware of NIPS’ possible consequences or risks, midwives balance their closeness and act within the boundaries of their resources and convictions. Closeness can prevent midwives from raising awareness to the level that, from a professional viewpoint, would be preferable to reach for fully informed decision making. Such midwifery abdication is a strategy opted for when internalized perceptions of midwifery practice and prioritization of the woman’s needs result in ‘knowing but failing to act’ [32]. In the constantly evolving maternity healthcare system, the ever-changing midwifery working field seems to affect midwives’ professional, proactive behaviour to the point that midwifery abdication is difficult to prevent [33]. Midwives’ tasks have multiplied without their available consultation time expanding accordingly, resulting in coping strategies such as falling back on fixed routines and existing protocols [34]; and this could include reducing NIPS counseling. 

For midwives who are convinced that conscious deliberation on NIPS risks and consequences is a prerequisite for every woman (and her partner), this can be frustrating, specifically if they fear routinization of NIPS procedures in women. If consultation opportunities are insufficient, or when values are stretched too far, midwives will adhere to informed consent as a baseline: providing formally required information and checking if decisions have been made. In cases where women avoid choices and routinely engage in NIPS, opting for this minimal level of informed consent is a means for midwives to respect women’s autonomy while staying close with them, irrespective of their own midwifery or personal values. In other words, negotiating women’s level of awareness allows midwives to stay true to their own professional value of empathetic nearness in protocolized contexts. Our study points to the need for reflection on the midwifery role in the maternity care system and a discussion of the essential midwifery values to be safeguarded within the medical care system. 

### 4.4. Strengths and Limitations

One major strength of this study is that upon data saturation we performed a saturation check and cross-validated our initial findings through additional interviews with two certified midwives who were highly experienced in midwifery and NIPS counseling and who had a broad, contemplative overview of Dutch midwifery practices. Another strength of this study is that the members of the team of researchers and authors are not personally involved in the Dutch maternity care system as midwives or as obstetricians. This lowers the risk of participants not wanting or daring to speak honestly during the interviews. Yet, this strength could also be considered a limitation, as the lack of an “insider” perspective in the team of researchers and authors may have compromised their understanding of the matters discussed by the midwives.

### 4.5. Implications for Practice

In an integrated maternity care system, an important challenge is to stay true to the core midwifery value of “being with women” while taking into account biomedical risk factors and helping women to make well-informed choices in line with their values. In the case of complex, value-laden decisions with potentially far-reaching consequences, such as NIPS, ensuring the right circumstances for women and their care providers to engage in an informed and deliberate decision-making process is critical. Dutch women consider a NIPS test as an additional test that they can use if they are willing to pay for it, and its acceptance is not morally obliged or a necessary aspect of responsible motherhood [35]. Notably, the Dutch government decided that as of 2023 pregnant women in the Netherlands who wish to use NIPS will no longer have to pay for it [36]. Instead, the full costs of NIPS will be covered by their basic healthcare insurance package, coverage that is compulsory for all Dutch citizens. This will remove one important barrier for well-informed NIPS decision making as we learned in our study. The cost-free availability of NIPS increases equitable access [37] but should not entail its standardized implementation in every pregnancy [38]. For that reason, counseling on NIPS remains a standard part of care offered to pregnant women in the Netherlands and will become an even more central aspect of Dutch midwifery counseling practices. Notably, the removal of the cost barrier can affect public attitudes and may increase societal pressure on women to choose NIPS. Full reimbursement of the cost of NIPS might result in an increase in women’s sense of responsibility to use testing [35], warranting sustained efforts by NIPS counselors to ensure freedom of choice for pregnant women [26]. One important implication of this impending policy change is that care providers who have been made responsible for NIPS counseling should be offered communicative training [39] as well as sufficient resources, such as room for proper counseling. Insufficient opportunity could result in involuntary economizing in counseling, preventing midwives from having a constructive dialogue that completely informs women and adequately helps them make a NIPS choice that reflects their values, circumstances, and needs. 

### 4.6. Directions for Future Research

Midwives are responsible for counseling on issues such as NIPS, place of birth, and pain medication, and face challenges that such counseling could pose to their own routines and values, in which physiological approaches are essential [40]. Therefore, counseling about NIPS and its implications should be an integral part of education for practicing and future midwives. Decision making is at the core of the translational process of bringing the use of genomics into reproductive and prenatal medicine; the full potential of NIPS can only be realized if women are able to make informed, value-based decisions about its use [41]. It is essential that future midwives take part in the reflection on midwifery values in the context of integrated care, which as of now essentially includes counseling about the option for NIPS. Learning to deal with such ethics-laden issues is a relatively novel task for midwives [42] and they need to cope with the moral distress such issues may evoke [43,44]. Follow-up studies should investigate how young midwives integrate and adapt their professional standards to the changing maternity care system, and what their specific needs in this context are. 

## 5. Conclusions

Providing information and guiding decisions about first-tier use of NIPS are still relatively new aspects in Dutch midwifery communication procedures. Dutch midwives develop novel routines in dealing with these topics within the constraints of their daily practice. In contrast to earlier expectations [14], Dutch midwives do not find NIPS counseling particularly easy. Rather, it may be a complex task, as both its timing and its goals may interfere with midwives’ core professional value of “being with women”, that is, being in empathetic nearness to women during their pregnancies and births. Raising women’s awareness about NIPS while respecting their autonomy sometimes compromises midwives’ professional core value which is rooted in trusting the natural process of pregnancy and birth and in empowering women to rely on their own capacities in this process. Balancing these traditional midwifery values and roles with NIPS counseling requires midwives to negotiate NIPS awareness in women. This is a particular challenge when women are ill informed, illiterate, hesitant, unwilling to receive information, or unable to understand the language. 

Dutch midwives attempt to find a workable equilibrium between their desire to remain close to the women they serve and to keep the professional distance needed for critical reflection and deliberation about choices. We recommend that midwifery students receive education and training in counseling about NIPS and its implications. In addition, future midwives, and indeed all midwives, should be invited to reflect on how NIPS counseling relates to their midwifery values and on how they can cope with the ethical and emotional aspects of counseling when practice and values come into conflict. 

## Data Availability

Not applicable.

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
