# Peer review of "Negotiating Awareness: Dutch Midwives’ Experiences of Noninvasive Prenatal Screening Counseling"

_ijerph, 2022, doi:10.3390/ijerph192215283_

Round 1

Reviewer 1 Report

Dear Editor,

thank you for giving me the opportunity to review this article on the role of midwifes in the prenatal counseling for the cell free DNA test. 

Despite I acknowledge the content of the manuscript to be of value for those countries where the prenatal counseling is held by midwifes, considering that the results may be applicable also to other professional figures of prenatal counseling, my major concern is that the number of the interviews (eight) is a bit to low to be representative of the whole population. 

Moreover, I find the description of the results a little too long and this makes you lose focus on the important results of the study. 

For this reason, I'm afraid that the article is not suitable for publication in this journal unless the authors are willing to increase the number of participants and to revise the manuscript accordingly to the suggestions.

Kind regards,

Author Response

1.The reviewer acknowledges the content of the manuscript to be of value for those countries where the prenatal counseling is done by midwives and points out that the results may be applicable to other professionals who provide prenatal counseling.

We thank the reviewer for this positive feedback.

2.The reviewer is concerned that the number of the interviews (eight) is a bit too low to be representative of the whole population.

The reviewer may have overlooked that we actually have 10 ten interviewees (see abstract, line 16, and section 2.1). We did two interviews as a saturation check and included them with the eight initial interviews. These two final interviews were conducted with senior midwives who had ample experience with midwifery counselling as well as with midwife coaching. We feel confident that our data represent a broad range of experiences. Please note that, as this is a qualitative study, representation of the whole population was, and could not be, its goal. Rather, its goal is to shed light of the various types of experiences that midwives have with NIPS counselling and the meanings they ascribe to those experiences.

3.The reviewer finds that the description of the results is a little too long, resulting in a loss of focus on the important results of the study.

We thank the reviewer for pointing this out. In accordance with the second reviewer’s comments, we restructured the results section. The themes are now numbered, and we inserted a subsection on barriers to women’s autonomous choice (within theme 2). In addition, we abbreviated some quotes by removing non-essential phrases, and omitting some parts of the results altogether that were, in hindsight, too elaborate, particularly fragments and quotes on information provided by telephone, and on awareness of the technicalities of test outcomes.

Reviewer 2 Report

This is a qualitative study of counselling pregnant individuals and their partners about non-invasive prenatal screening based on online in-depth interviews with 10 midwives in the Netherlands during a time of COVID-19 restrictions in the Netherlands.

MAJOR COMMENTS

1. Not entirely clear whether this is about first-tier use of NIPS, second-tier NIPS or both. I think it is first-tier, but this should be made clear

2. Recruitment of respondents was through personal networks of investigators, contact with midwifery practices, and referral from other participants. How many from each source? With this size of sample, it will probably not be possible to discern any pattern in response by source of recruitment. However, it is stated that the primary informants worked in group practice in Gelderland or Utrecht provinces. The former is large but is relatively not densely populated, whereas the latter is small and has high population density. This would be expected to induce variation by SES (and perhaps proportion of immigrants) in the populations served by the group practices. Further, a blended learning program for counsellors (including midwives) has been implemented in the Netherlands (PMID: 35499995) - might that have impacted on the variation among respondents? Given these caveats, a concern is that the sampling method is likely to have resulted in low variation among participants. 

3. I appreciate that conducting a study under COVID-19 restrictions is challenging. Do the investigators think that the study would have had different conclusions if focus-group discussions were implemented?

4. It would be helpful if the investigators checked the reporting of their study according to the COREQ criteria https://academic.oup.com/intqhc/article/19/6/349/1791966 

5. Was member checking considered?

6. Grounded theory approaches are generally aimed at theory development. However, theory development does not seem to have been pursued in this study.

7. In the conclusion, suggest highlighting existing barriers and providing recommendations based on the results.

OTHER COMMENTS

8. Did the authors consider use of computer software to assist their analysis? Is it possible that nuances may have been missed in the analysis?

9. Lines 190 to 197 refer to midwives’ beliefs rather than their awareness. Also, lines 241-4 seem to refer to pregnant women’s attitudes rather than knowledge. A new theme or sub-theme could be identified for this section.

10. Religious, language barrier or lower education levels could be categorized as a sub-theme of barriers to women’s autonomy.

11. Line 221 mentions "a good test result". Did the question of informing pregnant individuals up-front about the possibility of test failure (a substantial problem according to the Cochrane review by Badeau et al - PMID: 29125628) come up?

12. Unless I have missed it, I was surprised that there was no mention of the survey among Dutch midwives about their role as counsellors (PMID: 29024868).

Author Response

Reviewer 2.

The reviewer starts by summarizing that this is a qualitative study of counselling pregnant individuals and their partners about non-invasive prenatal screening based on online in-depth interviews with 10 midwives in the Netherlands during a time of COVID-19 restrictions in the Netherlands.

  1. Not entirely clear whether this is about first-tier use of NIPS, second-tier NIPS or both. I think it is first-tier, but this should be made clear.

We agree that this could be made more clear, and therefore we added in the abstract, in the introduction and in the conclusion that our study concerns a first-tier use of NIPS.

  1. Recruitment of respondents was through personal networks of investigators, contact with midwifery practices, and referral from other participants. How many from each source? With this size of sample, it will probably not be possible to discern any pattern in response by source of recruitment. However, it is stated that the primary informants worked in group practice in Gelderland or Utrecht provinces. The former is large but is relatively not densely populated, whereas the latter is small and has high population density. This would be expected to induce variation by SES (and perhaps proportion of immigrants) in the populations served by the group practices. Further, a blended learning program for counsellors (including midwives) has been implemented in the Netherlands (PMID: 35499995) - might that have impacted on the variation among respondents? Given these caveats, a concern is that the sampling method is likely to have resulted in low variation among participants.

We thank the reviewer for these thoughtful comments that have enabled us to elaborate on the recruitment of midwife respondents. We chose for recruitment throughout the larger areas of Gelderland and Utrecht, allowing us to cover both rural and urban contexts, traditionally catholic and traditionally protestant areas, and to include midwives from larger and smaller practices in various SES contexts. The two extra interviews were done with two midwives one of whom was stationed in the center of the Netherlands and one in the rural north of the Netherlands. In addition, as part of our purposive sampling, we made sure that all midwives who participated had received training in counselling for first-tier use of NIPS. This information was added to section 2.1.

  1. I appreciate that conducting a study under COVID-19 restrictions is challenging. Do the investigators think that the study would have had different conclusions if focus-group discussions were implemented?

Thanks for this check. Admittedly, talking via zoom can take some extra effort in listening and answering: interaction may at times be a little hampered by delays in transmission. However, the interviewers were used to the particularities of zoom and took some precautions, such as reserving extra time for the interviews and making sure to listen carefully for each answer and double checking to be certain they understood every nuance that the respondent gave.

  1. It would be helpful if the investigators checked the reporting of their study according to the COREQ criteria https://academic.oup.com/intqhc/article/19/6/349/1791966

We indeed used the consolidated criteria for reporting qualitative research (COREQ) to check the reporting of our study. In the revised version of our manuscript, this is indicated at the end of the Materials and Methods section.

  1. Was member checking considered?

Thanks for this question and for the opportunity to explain the method choices we made. For this study, member checking was not used. We used the clean language approach, producing data that reflect not so much the precise answers to specific questions (that participants could have checked) but rather communicative ways of expressing thoughts, ideas and experiences in a certain thematic field. This type of data is rather less suited for back checking with participants. Nevertheless, all participants were invited to contact the interviewer if anything came up that they would like to add to what they had said in the interview, or if in hindsight they wished to withdraw anything they had said during the interview. None of the participants made use of this opportunity.

  1. Grounded theory approaches are generally aimed at theory development. However, theory development does not seem to have been pursued in this study.

We thank the reviewer for pointing out this important matter. We now provide further explanation of our starting point in literature. We included two references in our introduction that describe the experience with, and training for, NIPS counseling in Dutch midwives. Subsequently and following up on these references, we explore the kinds of experiences and reflections Dutch midwives have regarding NIPS counseling. This resulted in the following extra sentences at the end of our introduction:

Dutch midwives were found to give information about NIPS and to be confident with their abilities to counsel about NIPS (Martin et al. 2018); since the national implementation of NIPS, Dutch midwives have been trained to counsel pregnant women about first-tier use of this type of test (Martin et al., 2022). Anticipating the nationwide implementation of NIPS, Dutch midwives expected that counseling for NIPS as first screening test would become more important, and that it would be relatively easy (Martin et al., 2018). An important question is therefore, what kinds of experiences and reflections Dutch midwives actually have regarding  counselling about NIPS.

  1. 7. In the conclusion, suggest highlighting existing barriers and providing recommendations based on the results.

Thanks for this invitation. We now add that in contrast to earlier expectations, Dutch midwives find NIPS counselling not particularly easy; we describe barriers to effective counselling and a suggestion at the end of the conclusion section:

This is a particular challenge when women are ill informed, illiterate, hesitant, unwilling to receive information, or unable to understand the language.

We recommend that midwifery students receive education and training in counselling about NIPS and its implications. In addition, future midwives, and indeed all midwives, should be invited to reflect on how NIPS counseling relates to their midwifery values and on how they can cope with the ethical and emotional aspects of counselling when practice and values come into conflict.

  1. Did the authors consider use of computer software to assist their analysis? Is it possible that nuances may have been missed in the analysis?

Thanks for the opportunity to explain our analytic strategy.  We used Atlas-TI and now point this out in section 2.3 (Analyses).

  1. a Lines 190 to 197 refer to midwives’ beliefs rather than their awareness.

Thanks to the reviewer for pointing this out. We now see that this sentence is unclear. What we intended to express is that women should be aware (etc). The theme here really is awareness. The text was altered accordingly.

9.b Also, lines 241-4 seem to refer to pregnant women’s attitudes rather than knowledge. A new theme or sub-theme could be identified for this section.

Thanks to the reviewer for pointing this out. This sentence is unclear. We intended to explain that women do not want to know more about their options, fitting with the knowledge theme. The text was altered accordingly.

  1. Religious, language barriers, or lower education levels could be categorized as a sub-theme of barriers to women’s autonomy.

We thank the reviewer for this valuable suggestion, which we followed by identifying a subsection with the suggested heading.

  1. 11. Line 221 mentions "a good test result". Did the question of informing pregnant individuals up-front about the possibility of test failure (a substantial problem according to the Cochrane review by Badeau et al - PMID: 29125628) come up?

Thanks to the reviewer for pointing this out. Our intent was to explain that women, in the experience of midwives, often expect that they will receive good news as test result. We changed the text accordingly. The possibility of a test failure was not explicitly mentioned.

  1. Unless I have missed it, I was surprised that there was no mention of the survey among Dutch midwives about their role as counsellors (PMID: 29024868).

We regret that we failed to mention this relevant reference. We have inserted it in the introduction.

Round 2

Reviewer 1 Report

Dear Authors,

thank you for revising the manuscript.

I think it is suitable for publication. 

Kind regards,

Author Response

We thank Reviewer 1 for considering our revised manuscript and we are happy to hear that Reviewer 1 thinks the manuscript is now suitable for publication.

Kind regards, also on behalf of all co-authors,

Marieke de Vries

Reviewer 2 Report

Appreciate responses to previous comments, and changes made. 

1. lines 57-59 refer to national implementation of NIPS, citing Martin et al., 2022. What is unclear to me is whether this national implementation was in the context of the TRIDENT research initiative, or had become implemented as routine practice. I suggest that you make this clear

2. In response to previous comment 2, you responded "In addition, as part of our purposive sampling, we made sure that all midwives who participated had received training in counselling for first-tier use of NIPS". I think this part of your explanation was not added to section 2.1. I think it should be, especially in context of first comment this time round.

3. Thank you for clarifying that you used the COREQ reporting guideline. Please include completed checklist as appendix

4. In response to the comment about member checking, you state that you gave respondents the opportunity to add to or withdraw anything stated in interview, but that none had made use of the opportunity. You should say so.

Author Response

We thank Reviewer 2 for the helpful additional comments and suggestions. Below, we provide a point-by-point response indicating how we have incorporated these in the revision of our manuscript. We believe the manuscript has improved as a result.

Kind regards, also on behalf of all co-authors,

Marieke de Vries

1. lines 57-59 refer to national implementation of NIPS, citing Martin et al., 2022. What is unclear to me is whether this national implementation was in the context of the TRIDENT research initiative, or had become implemented as routine practice. I suggest that you make this clear --> CLARIFIED

2. In response to previous comment 2, you responded "In addition, as part of our purposive sampling, we made sure that all midwives who participated had received training in counselling for first-tier use of NIPS". I think this part of your explanation was not added to section 2.1. I think it should be, especially in context of first comment this time round. --> REPAIRED

3. Thank you for clarifying that you used the COREQ reporting guideline. Please include completed checklist as appendix --> CHECKLIST INCLUDED IN APPENDIX

4. In response to the comment about member checking, you state that you gave respondents the opportunity to add to or withdraw anything stated in interview, but that none had made use of the opportunity. You should say so. --> DONE